# Liposomal TLR9 Agonist Combined with TLR2 Agonist-Fused Antigen Can Modulate Tumor Microenvironment through Dendritic Cells

**DOI:** 10.3390/cancers12040810

**Published:** 2020-03-28

**Authors:** Kuan-Yin Shen, Hsin-Yu Liu, Wan-Lun Yan, Chiao-Chieh Wu, Ming-Hui Lee, Chih-Hsing Leng, Shih-Jen Liu

**Affiliations:** 1National Institute of Infectious Diseases and Vaccinology, National Health Research Institutes, Miaoli 350401, Taiwan; shenky057@nhri.edu.tw (K.-Y.S.); isbelliu@gmail.com (H.-Y.L.); 044027@nhri.edu.tw (W.-L.Y.); 050818@nhri.edu.tw (C.-C.W.); 011013@nhri.edu.tw (M.-H.L.); leo_leng@adimmune.com.tw (C.-H.L.); 2Graduate Institute of Life Sciences, National Defense Medical Center, Taipei 114201, Taiwan; 3Graduate Institute of Biomedical Sciences, China Medical University, Taichung 404394, Taiwan; 4Graduate Institute of Medicine, Kaohsiung Medical University, Kaohsiung 807378, Taiwan

**Keywords:** lipoprotein, liposome, DOTAP, Toll-like receptor 2, phosphodiester CpG

## Abstract

Dendritic cells (DCs) are antigen-presenting cells involved in T cell activation and differentiation to regulate immune responses. Lipoimmunogens can be developed as pharmaceutical lipoproteins for cancer immunotherapy to target DCs via toll-like receptor 2 (TLR2) signaling. Previously, we constructed a lipoimmunogen, a lipidated human papillomavirus (HPV) E7 inactive mutant (rlipoE7m), to inhibit the growth of HPV16 E7-expressing tumor cells in a murine model. Moreover, this antitumor effect could be enhanced by a combinatory treatment with CpG oligodeoxynucleotides (ODN). To improve safety, we developed a rlipoE7m plus DOTAP liposome-encapsulated native phosphodiester CpG (POCpG/DOTAP) treatment to target DCs to enhance antitumor immunity. We optimized the formulation of rlipoE7m and POCpG/DOTAP liposomes to promote conventional DC and plasmacytoid DC maturation in vitro and in vivo. Combination of rlipoE7m plus POCpG/DOTAP could activate conventional DCs and plasmacytoid DCs to augment IL-12 production to promote antitumor responses by intravenous injection. In addition, the combination of rlipoE7m plus POCpG/DOTAP could elicit robust cytotoxic T lymphocytes (CTLs) by intravenous immunization. Interestingly, the combination of rlipoE7m plus POCpG/DOTAP could efficiently inhibit tumor growth via intravenous immunization. Moreover, rlipoE7m plus POCpG/DOTAP combined reduced the number of tumor-infiltrating regulatory T cells dramatically due to downregulation of IL-10 production by DCs. These results showed that the combination of rlipoE7m plus POCpG/DOTAP could target DCs via intravenous delivery to enhance antitumor immunity and reduce the number of immunosuppressive cells in the tumor microenvironment.

## 1. Introduction

Overcoming immunosuppression by the tumor microenvironment is a difficult task for cancer immunotherapy [1]. Triggering immunostimulatory signaling and blocking immune checkpoints are two major routes to enhance antitumor immunity [2,3,4]. Dendritic cells (DCs) are important for the regulation of immunological stimulation and tolerance [5,6]. DCs are capable of gathering and processing antigens to present immunogenic epitopes for cytotoxic T lymphocyte (CTL) activation against tumor cells and virus-infected cells [7]. In addition, DC maturation is promoted by pathogen-associated molecular pattern (PAMP) stimulation of pathogen recognition receptors (PRRs), such as toll-like receptors (TLRs) and nucleotide-binding oligomerization domain (NOD)-like receptors (NLRs) [8,9]. Thus, the adjuvanticity of PAMP-derived immunostimulators can promote tumor-associated antigen-induced antitumor immunity [10]. Multiple TLR agonist combinations can induce robust antitumor immunity and overcome the immunosuppression by regulatory T cells (Tregs) [11]. In our previous study, a TLR2 agonist recombinant lipoprotein combined with the TLR9 agonist CpG oligonucleotide (CpG ODN) could dramatically enhance antitumor efficacy and reduce the number of tumor-associated macrophages (TAMs) and myeloid-derived suppressor cells (MDSCs) [12]. In addition, a recombinant lipoprotein could increase the uptake of 1 2-Dioleoyloxy-3-trimethylammonium propane (DOTAP) liposomes by DCs to promote antitumor immunity [13]. Thus, DC-targeting lipoprotein-coated DOTAP liposomes are a promising approach for TLR agonist and antigen delivery to induce efficient anticancer immunotherapy.

Lipoimmunogens have been developed to enhance the immunogenicity of antigens and promote humoral and cellular immunity [14,15]. Lipoproteins with lipid moieties at the conserved lipid box sequence are derived from the bacterial outer membrane and, as TLR2 agonists, are immunostimulators. In addition, recombinant lipoproteins have been constructed from lipidated pathogen-associated antigens against bacteria, viruses, or malignant tumors [16,17,18]. Recombinant lipoproteins with a tri-acyl lipid moiety could promote dendritic cell (DC) maturation to induce secretion of inflammatory cytokines, such as tumor necrosis factor-α (TNF-α), interleukin (IL)-6, and IL-12, and upregulation of costimulatory molecules, including CD40, CD80, and CD86 through TLR2 [16]. Recombinant lipoproteins are molecular vaccines with intrinsic self-adjuvant activity.

According to our previous studies, the recombinant lipidated tumor-associated antigen derived from a human papillomavirus (HPV) E7 inactive mutant (rlipoE7m) combined with the TLR9 ligand phosphorothioate CpG ODN (PSCpG) can dramatically induce tumor regression [12]. The recombinant lipoprotein combined with PSCpG immunization not only elicited CTL responses but also reduced the number of tumor-infiltrating immunosuppressive cells M2-like TAMs, MDSCs, and Tregs [19]. Therefore, TLR2 and TLR9 signaling can synergistically induce robust CTLs and modify the tumor microenvironment toward regression [20]. However, PSCpG is more nuclease-resistant than phosphodiester CpG ODN (POCpG) and its stability resulted in some side effects in mice, such as arthritis induction and splenomegaly [21,22]. To preserve the adjuvant activity of CpG ODN and avoid the side effect of PSCpG, liposome-encapsulated POCpG has been investigated and has been shown to control infectious diseases and malignant tumors [23].

Cationic liposomes can encapsulate anionic materials, such as DNA, to form a stable complex by static electronic interaction for delivery. The cationic liposome DOTAP can deliver DNA efficiently to cells [24]. DOTAP can also improve the antitumor immunogenicity of proteins and synthetic peptides derived from tumor-associated antigens. DOTAP can activate bone marrow-derived dendritic cells (BMDCs) to secrete IL-12 and C-C motif chemokine ligand 2 (CCL2) via the reactive oxygen species (ROS) and mitogen-activated protein kinase (MAPK) p38 pathways [25]. In our previous investigation, rlipoE7m and DOTAP could form a stable complex to slow antigen release to target DCs and enhance the therapeutic effect in the TC-1 tumor model [13]. In this study, we further demonstrated that the rlipoE7m plus DOTAP-encapsulated POCpG formulation can target DCs via intravenous administration and improve antitumor responses. Furthermore, we also investigated DC maturation in vivo and the changes in the tumor microenvironment.

## 2. Results

### 2.1. Formulation of Lipoprotein and Phosphodiester CpG within Cationic Liposomes

To combine the lipoprotein POCpG and DOTAP liposomes, we first optimized the dosage of rlipoE7m and POCpG with DOTAP liposomes. A total of 2 × 10^6^ BMDCs were treated with the indicated concentrations of rlipoE7m and POCpG with DOTAP liposomes for 18 h, and then cytokine concentrations in the supernatant were determined by ELISA. The production of proinflammatory cytokines IL-1β, IL-6, IL-12p70, and TNF-α by the treated BMDCs was dose-dependent (Figure 1A). Interestingly, 10 μg of rlipoE7m plus POCpG/DOTAP combined could stimulate IL-1β production by BMDCs significantly better than 2 μg of rlipoE7m plus POCpG/DOTAP combined. The results indicate that the recombinant lipoprotein was the major factor triggering IL-1β production. Next, we determined the encapsulation rate of POCpG/DOTAP. In the preparation step of the lipid film suspension, 2 μg or 10 μg of POCpG were added to the lipid film for encapsulation and then free CpG in the supernatant was measured. As shown in Figure 1B, 100 nmole DOTAP could encapsulate 2 μg and 10 μg POCpG at a 99% encapsulation rate. Anionic POCpG encapsulated by cationic DOTAP exhibited an efficient encapsulation rate due to the electrostatic interaction between DOTAP and POCpG. We further examined the adsorption rate of 10 μg rlipoE7m with 100 nmole DOTAP liposomal formulation containing 2 μg or 10 μg POCpG. We used an ELISA approach to measure the amount of rlipoE7m adsorbed with the liposomal combination. The adsorption rate of 10 μg rlipoE7m with 100 nmole DOTAP liposomes containing 0, 2, or 10 μg POCpG was approximately 70% (Figure 1C). Although 100 nmole DOTAP had already encapsulated 10 μg POCpG, it could still adsorb 70% of 10 μg rlipoE7m as well as 100 nmole DOTAP only. POCpG did not influence the abortion of rlipoE7m by 100 nmole DOTAP liposomes. Thus, we determined the liposomal combination of recombinant lipoprotein and POCpG to contain 10 μg of rlipoE7m and 10 μg of POCpG with 100 nmole of DOTAP liposomes. In a previous study, we demonstrated that the rlipoE7m DOTAP liposomal combination could efficiently target DCs in draining lymph nodes [13]. In this study, we determined whether the combination treatment could target vascular DCs. We cultured bone marrow-derived plasmacytoid DCs (pDCs) that were incubated with rlipoE7m plus POCpG/DOTAP combined for 18 h for cytokine detection in the supernatant. The results showed that POCpG/DOTAP could effectively stimulate IFN-α, IL-6, and IL-12p70 production by pDCs compared with POCpG without DOTAP encapsulation. Moreover, rlipoE7m plus POCpG/DOTAP combined significantly enhanced IFN-α, IL-6, and IL-12p70 production compared with rlipoE7m or POCpG/DOTAP (Figure 1D). We developed a stable composition of a TLR2 agonist on the DOTAP liposome surface and encapsulated a TLR9 agonist in the DOTAP liposome to synergistically trigger TLR2 and TLR9 pathways for DC activation.

Costimulatory molecules such as CD40, CD80, CD83, and CD86 expressed on DCs play important roles in inducing cytotoxic T lymphocyte (CTL) activation. Next, to assess the expression of costimulatory molecules on BMDCs after rlipoE7m plus POCpG/DOTAP combination treatment, BMDCs were analyzed by flow cytometry. As shown in Figure 2A, rlipoE7m plus POCpG/DOTAP combined significantly upregulated CD40 (267.8 ± 15.3%), CD80 (132.6 ± 6.7%), CD83 (170.3 ± 23.6%), and CD86 (167.5 ± 35.2%) expression compared with the control group. Interestingly, 100 nmole DOTAP liposomes without TLR agonists could upregulate CD40 (160.2 ± 30.4%) and CD83 (170.2 ± 43.7%). This observation was consistent with our previous study [13]. Next, we further investigated whether rlipoE7m plus POCpG/DOTAP combined could target DCs to promote maturation via intravenous immunization. C57BL/6 mice were immunized with rlipoE7m, rlipoE7m plus CpG, or rlipoE7m plus POCpG/DOTAP. Eighteen hours later, the maturation markers CD40, CD80, CD83, and CD86 of CD11c^+^ conventional DCs (cDC) and PDCA1^+^ pDCs from spleens were analyzed by flow cytometry. The results showed that rlipoE7m plus POCpG/DOTAP combined could significantly upregulate these maturation markers on cDCs and pDCs compared with treatment without DOTAP liposomes (Figure 2B,C). This result demonstrated that DOTAP liposomes could facilitate rlipoE7m and POCpG codelivery to target cDCs and pDCs in the vascular system. In addition, we characterized the cytokine profiles of pDCs stimulated by rlipoE7m plus POCpG/DOTAP combined. POCpG without DOTAP encapsulation did not effectively promote costimulatory molecule expression on BMDCs compared with POCpG/DOTAP. Because POCpG can be digested by nuclease in vivo, we used DOTAP-encapsulated POCpG to prolong its half-life. In addition, the negatively charged rlipoE7m could be easily adsorbed by the positively charged POCpG/DOTAP liposomes to form immunostimulatory complexes. This results indicated that rlipoE7m plus POCpG/DOTAP combined could target pDC to enhance IL-12p70 production to promote the Th1 response for the development of antitumor immunity. In summary, rlipoE7m plus POCpG/DOTAP combined could upregulate the production of the Th1-biased cytokine IL-12p70 and the expression of costimulatory molecules on DCs to provide appropriate signals for CTL activation.

### 2.2. The Combination of rlipoE7m and POCpG/DOTAP Enhanced CTL Responses

To further investigate whether rlipoE7m plus POCpG/DOTAP combined could be delivered efficiently to enhance CTL responses, C57BL/6 mice were immunized with rlipoE7m, rlipoE7m plus POCpG, or the liposomal combination of rlipoE7m plus POCpG/DOTAP at days 0 and 7 via intravenous injection. At day 14, cells were collected from spleens, treated with RAH peptide for 48 h and restimulated to induce IFN-γ production by antigen-specific T cells. Immunization with rlipoE7m plus POCpG/DOTAP combined elicited IFN-γ-producing T cells (161 ± 7.5 spots) more efficiently than rlipoE7m immunization (20 ± 2.2 spots) or rlipoE7m plus POCpG immunization (30 ± 6.5 spots) via intravenous injection (Figure 3A). Furthermore, we analyzed the HPV16 E7-specific CTL population in the spleens of immunized mice. The HPV16 E7-specific CTL population was determined by the RAH tetramer and CD8 double-positive staining. The results showed that rlipoE7m plus POCpG/DOTAP combined immunization could elicit robust HPV16 E7-specific CTL populations (0.113 ± 0.011% of CD8 T cells) compared with rlipoE7m plus POCpG immunization (0.067 ± 0.002% of CD8 T cells) (Figure 3B). Overall, DOTAP liposomes efficiently delivered rlipoE7m and POCpG to induce a strong CTL response via intravenous immunization.

### 2.3. The Combination of rlipoE7m and POCpG/DOTAP Could Induce Tumor Regression

We have demonstrated that immunization with rlipoE7m combined with PSCpG could induce large tumor regression [12]. In this study, we used a large tumor model in which TC-1 grew to ~80 mm^3^ at 14 days after tumor inoculation. To assess the therapeutic effect of the combination of rlipoE7m plus POCpG/DOTAP, TC-1 tumor-bearing mice were immunized via intravenous injection at day 14 and then boosted at day 21. The results showed that one immunization with rlipoE7m plus POCpG/DOTAP combined could delay tumor growth (1005 ± 301 mm^3^ at day 33) as well as rlipoE7m plus POCpG immunization (1242 ± 341 mm^3^ at day 33) compared with the mock control (2222 ± 395 mm^3^ at day 33) (Figure 4A). However, two immunizations with rlipoE7m plus POCpG/DOTAP combined via intravenous injection could inhibit tumor growth (22 ± 14 mm^3^ at day 33), even eradicating the tumors, compared with rlipoE7m plus POCpG immunization (1292 ± 334 mm^3^ at day 33). (Figure 4B). Surprisingly, two immunizations with rlipoE7m plus POCpG/DOTAP combined could eliminate 80% of tumors between days 27 to 37 via intravenous injection (Figure 4C). Overall, two immunizations with rlipoE7m plus POCpG/DOTAP combined could induce antitumor immunity to eradicate TC-1 tumors compared with rlipoE7m or rlipoE7m plus POCpG immunization.

### 2.4. The Combination of rlipoE7m and POCpG/DOTAP Could Reduce the Number of regulatory T Cells in Tumors

Tumor-associated immunosuppressive cells, Tregs, and MDSCs affect the efficacy of immunotherapy. We further investigated whether rlipoE7m plus POCpG/DOTAP combined could influence tumor-infiltrating cells, including Tregs and MDSCs, to alter the tumor microenvironment. TC-1 tumor-bearing mice were immunized on days 14 and 21, and tumor-infiltrating cells were collected from the tumors 3 and 5 days after final immunization. Although some tumors had shrunk after rlipoE7m plus POCpG/DOTAP combined immunization, we still collected data from 5 mice in this group. The results showed that rlipoE7m plus POCpG/DOTAP combined could increase the CD8^+^ T cell population (day 3: 10.53 ± 1.13%; day 5: 15.45 ± 2.22%) in tumors compared with rlipoE7m plus POCpG (day 3: 4.94 ± 0.91%; day 5: 7.98 ± 1.53%) at day 5 (Figure 5A). Although the tumor-infiltrating CD4^+^ T cell population was not different after immunization (Figure 5B), rlipoE7m plus POCpG/DOTAP combined significantly decreased the number of tumor-infiltrating Tregs (day 3: 15.82 ± 6.24%; day 5: 19.42 ± 4.65%) compared with rlipoE7m plus POCpG (day 3: 30.28 ± 4.03%; day 5: 36.33 ± 1.44%) (Figure 5C). In addition, rlipoE7m plus POCpG/DOTAP combined did not influence the population of splenic Tregs (Figure 5D). Interestingly, rlipoE7m plus POCpG/DOTAP combined also reduced CD11b^+^ Gr-1^lo^ monocytic MDSCs (16.59 ± 2.11%) compared with the mock control (37.94 ± 9.08%) on day 5 after immunization (Figure 5E). In contrast, rlipoE7m plus POCpG/DOTAP combined did not reduce CD11b^+^ Gr-1^hi^ granulocytic MDSCs (Figure 5F). The results demonstrated that rlipoE7m plus POCpG/DOTAP combined not only elicited robust CTLs but also reduced the number of immunosuppressive cells, Tregs, and MDSCs via intravenous immunization to alter the immunosuppressive tumor microenvironment.

IL-10 is a critical immunosuppressive cytokine that regulates immunosuppressive cell activation. We evaluated IL-10 production by BMDCs treated with rlipoE7m plus POCpG/DOTAP combined. Two kinds of DCs, cDCs and pDCs, were cultured from bone marrow cells and then treated with rlipoE7m, CpG, POCpG/DOTAP, or rlipoE7m plus POCpG/DOTAP for 24 h. IL-10 in the supernatant was determined by ELISA. Interestingly, rlipoE7m plus POCpG/DOTAP-treated cDCs could secrete less IL-10 than rlipoE7m- or rlipoE7m plus POCpG-treated cDCs (Figure 6A). In addition, rlipoE7m plus POCpG/DOTAP-treated cDCs also secreted less IL-10 than rlipoE7m- or rlipoE7m plus POCpG-treated pDCs (Figure 6B). In conclusion, rlipoE7m plus POCpG/DOTAP combined could increase IL12p70 to promote Th1 responses and reduce IL-10 production to diminish immunosuppressive cells and alter the tumor microenvironment via intravenous immunization.

## 3. Discussion

Based on this study, using DOTAP liposomes to carry lipoproteins and POCpG is a feasible approach to develop biocompatible and biodegradable cancer immunotherapeutic vaccines. Encapsulating POCpG in DOTAP can not only prolong the half-life but also increase the delivery efficiency to stimulate TLR9, which is located at intracellular endosomes [26]. Because DOTAP can activate DCs via the ROS pathway, the ROS pathway may play a role in regulating the TLR2 and TLR9 pathways. Because the ROS pathway can upregulate IL-1β production [27], we suggested that the ROS pathway may be involved in the induction of IL-1β production by the rlipoE7m plus POCpG/DOTAP combined treatment. Moreover, IL-1β can promote DC maturation and enhance the secretion of IL-12 to induce a Th1-biased immune response [28,29]. In addition, we had demonstrated that lipoprotein and DOTAP can upregulate CD83 expression though TLR2 and ROS signaling [13]. The combination of rlipoE7m and POCpG/DOTAP also upregulated the costimulatory molecule CD83 as did rlipoE7m plus DOTAP combined. Therefore, we suggested that the ROS and TLR2 pathways were required for CD83 upregulation, but the TLR9 pathway was not. We also observed that rlipoE7m and POCpG without DOTAP could not activate cDCs and pDCs via intravenous administration. These results indicate that DOTAP liposomes are necessary for rlipoE7m and POCpG delivery via intravenous administration.

TLR2 and TLR9 agonists can facilitate the antigen cross-presentation pathway to efficiently promote CTL priming [30,31,32]. TLR2 and TLR9 agonist codelivery in combination with DOTAP liposomes may also enhance antigen presentation for CTL priming. TLR agonists can have direct impact on the Treg population. TLR9 can downregulate Treg suppression more than TLR2 ligands can [33]. In addition, TLR9 agonists inhibit the suppressive activity of Tregs [34,35]. Furthermore, mature DCs expressing high levels of CD86 and CD80 can modulate the suppressive function of Tregs in T cell activation [36]. In contrast, CD83-expressing mature DCs and Foxp3^+^ Tregs are two opposing factors of anticancer immunity [37]. Consequently, our approach of a DOTAP liposomal combination is successful in targeting DCs to increase CTLs and decrease Tregs to promote antitumor efficacy.

In this study, we analyzed the DOTAP liposomal combination for lipoprotein and POCpG codelivery via an intravenous route to target cDCs and pDCs to promote antitumor responses. DOTAP liposomes not only successfully combined TLR2 and TLR9 agonists to target DCs but also altered the cytokine production triggered by TLR2 and TLR9 agonists triggering cytokine production, such as IL-1β production and IL-10 reduction. IL-10 plays multiple roles for immunosuppression in the tumor microenvironment. IL-10 may promote Gr-1-highly expressing cells to differentiate into TAMs, and TAMs further produce IL-10 for immune suppression [38]. IL-10 can lead STAT3- and SOCS3-mediated suppression to inhibit TLR and IFN-γ signaling pathways [39]. Moreover, IL-10 can inhibit the antigen presentation ability of DCs for T cell activation [40]. Although TLR agonists can induce IL-10, which can negatively regulate the MyD88-dependent pathway [41], rlipoE7m plus POCpG/DOTAP combined treatment triggers low IL-10 production of cDCs and pDCs to promote antitumor immunity. We suggest that DOTAP influences TLR2 and TLR9 signaling-mediated IL-10 production in cDCs and pDCs. Thus, the DOTAP liposome is not only a carrier for lipoprotein and POCpG delivery, but also alters the TLR signaling pathway toward Th1 immune responses.

Liposomes are an effective delivery system for nano drugs and nucleic acids in clinical applications. The cationic DOTAP liposome can adsorb or encapsulate RNA and DNA for genetic transfection [42]. Liposomes are a feasible approach for intravenous and intramuscular injection into circulation system to induce humoral and cellular immunity [43,44,45]. To turn cold tumors into hot tumors, immunotherapeutic agents should be delivered into the tumor site to alter the immunosuppressive microenvironment [46,47]. In this report, we demonstrated that the DOTAP liposome could carry TLR2-fused antigens and TLR9 agonists to enhance DCs activation and reduce Tregs in the tumor microenvironment. Our study could provide some ideas for the design of delivery mechanisms for immunostimulators or modulators for cancer immunotherapy through regulation of the tumor microenvironment.

## 4. Materials and Methods

### 4.1. Recombinant Lipoprotein Preparation

The recombinant lipoimmunogen, rlipoE7m, is an HPV 16 E7 inactive mutant that was cloned in pET-22b with a hexahistidine tag for expression and purification [16]. Briefly, rlipoE7m was purified by disrupting the harvested cells in a French press (Constant Systems, Daventry, UK) at 27 Kpsi in a homogenization buffer (50 mM Tris, pH 8.9). Solubilization buffer (1% Triton X-100; 50 mM Tris, pH 8.9) was used to solubilize rlipoE7m from the cell lysate pellet after 100,000 g centrifugation. The supernatant was incubated with 5 mL of Ni-NTA resin (Qiagen, San Diego, CA, USA) overnight and then loaded into a column (1.6 cm i.d. × 2.5 cm). The column was washed with the solubilization buffer, and rlipoE7m was eluted with 100 mM imidazole-containing solubilization buffer. To remove the LPS endotoxin, the purified rlipoE7m was bound to Chelating Sepharose resin (GE Healthcare, Waukesha, WI, USA) coupled with copper, and extensively washed with solubilization buffer. The solubilization buffer of endotoxin-free rlipoE7m was exchanged with PBS by dialysis. Finally, LPS levels were determined by LAL test (Associates of Cape Cod, Inc, East Falmouth, MA.) and found to be below 0.003 EU/mg. The POCpG sequence is 5’- TCCATGACGTTCCTGACGTT -3’ (GeneDireX, NV, USA).

### 4.2. Liposome Preparation

We used the lipid film method to prepare DOTAP liposomes. For this purpose, 1,2-dioleoyl-3-trimethylammonium-propane (DOTAP, chloride salt) was purchased from Avanti and loaded into cell culture-grade tubes (Corning, NY, USA). DOTAP chloride salt was evaporated to a chloroform solution in lipid films and then vacuumed dry overnight at room temperature. Lipid films were hydrated for 12 h by adding the required amount of water to a final concentration of 10 mg/mL. The suspensions were sonicated in a bath-type sonicator for 10 min at 50 °C and then extruded by the mini-extruder (Avanti, Alabaster, AL, USA) through 400, 200, and 100 nm membrane filters sequentially. The DOTAP liposomes were stored at 4 °C before use. To encapsulate POCpG in 100 nmole DOTAP, 2 μg or 10 μg POCpG in 50 µL of ddH_2_O were added to the DOTAP lipid film followed by the procedure for DOTAP liposome preparation.

### 4.3. Encapsulation Rate and Adsorption Rate Determination

After the encapsulation process of POCpG in DOTAP, 2 μg phosphodiester CpG/DOTAP or 10 μg phosphodiester CpG/DOTAP were loaded into 300 KDa Vivaspin 500 Centrifugal Concentrators (Sartorius) with 13,000 *g* centrifugation for 2 min. The flow-through was collected, and then the concentration of nucleotide was measured at OD260 nm by Nanodrop 2000c (Thermo Scientific). The encapsulation rate was calculated using the following formula: (Encapsulation rate (%) = (Concentration of original–Concentration of flow-through)/Concentration of original) × 100%.

To determine the rlipoE7m adsorption by POCpG/DOTAP liposomes, 10 μg of rlipoE7m was admixed with 2 μg POCpG/DOTAP or 10 μg POCpG/DOTAP for 10 mins. A total of 100 μL admixtures were loaded into a 100-KDa Vivaspin 500 microconcentrator with 13,000 *g* centrifugation for 4 min. The concentration of rlipoE7m in the flow-through was determined with sandwich ELISA. Briefly, the flow-through was incubated in a rabbit anti-His tag antibody (LTK BioLaboratories)-coated plate. Mouse anti-E7 antibody (CERVIMAX) detected the captured E7 protein. To develop the ELISA, HRP-conjugated goat antimouse antibody and TMB substrate were added to the plates. The intensity of the colored product was directly proportional to the concentration of rlipoE7m. The adsorption rate was calculated by the following formula: (adsorption rate (%) = (concentration of original–concentration of flow-through)/ concentration of original) × 100%.

### 4.4. Analysis of the Expression of Costimulatory Molecules on BMDCs and Plasmacytoid DCs

BMDCs cultured from the bone marrow of C57BL/6 mice were assessed as previously described [30]. Briefly, bone marrow cells were cultured at a density of 2 × 10^6^ cells/10 mL in petri dishes with complete RPMI-1640 medium (10% FBS, 10% penicillin/streptomycin, 0.05 μΜ beta-mercaptoethanol, 0.025 M HEPES, 1 μΜ sodium pyruvate) containing 20 ng/mL recombinant mouse granulocyte-macrophage colony-stimulating factor (GM-CSF) (PeproTech) for DC differentiation. On day 3, an additional 10 mL of complete RPMI medium containing 20 ng/mL GM-CSF was added. On day 6, the cells were collected from each dish and then washed with complete RPMI-1640 medium. To test the bioactivity of rlipoE7m combined with POCpG/DOTAP liposome, BMDCs (1 × 10^6^ cells/mL) were stimulated with the indicated concentrations of rlipoE7m alone or combined with POCpG/DOTAP liposomes for 18 h. In addition, 2 × 10^6^/20 mL bone marrow cells were cultured with RPMI-10 containing 100 ng/mL Flt3 ligand for 9 days to derive plasmacytoid DCs. Cells were not disturbed until collection. Nine days later, suspension cells were collected and counted for experiments. The expression of costimulatory molecules on BMDCs or pDCs was determined by staining with anti-PDCA1 (129C), anti-CD40, anti-CD80, anti-CD83, and anti-CD86 fluorescent antibodies (clones 3/23, 16-10A1, Michel-1F, and GL-1, respectively, eBioscience, San Diego, CA, USA,) and then analyzed by flow cytometry (FACSCalibur, BD bioscience, San Jose, CA, USA). The IL-1β, IL-6, IL-12p70, IFN-α, and TNF-α secretion from BMDCs or pDCs was determined by using ELISA kits (eBioscience).

### 4.5. Analysis of IFN-γ-Producing CTLs

C57BL/6 mice were given intravenous injections of DOTAP, 10 μg rlipoE7m, 10 μg rlipoE7m combined with 10 μg POCpG, or 10 μg rlipoE7m combined with 10 μg POCpG/100nmole DOTAP on days 0 and 7. Fourteen days after the first immunization, the mice were sacrificed, and then 5 × 10^5^ splenocytes were restimulated with 5 μg/mL RAH, the peptide derived from the HPV16 E7 murine CTL epitope, for 48 h. The number of IFN-γ-secreting cells after restimulation was determined by ELISPOT (BD Biosciences, San Jose, CA, USA).

### 4.6. Animal and Tumor Model

C57BL/6JNlac mice were purchased from the National Laboratory Animal Center, Taiwan. All experimental mice were held in a pathogen-free environment at the Laboratory Animal Center of the National Health Research Institutes (NHRI). The animals were used in compliance with institutional animal health care regulations, and all animal experimental protocols were approved by the NHRI IACUC (approval ID: NHRI IACUC-104028). The TC-1 cell line, which expresses HPVE6 and E7 mouse epithelial lung cancer, was a kind gift from Dr. T-C. Wu (Johns Hopkins University, Baltimore, MD, USA). To assess therapeutic antitumor effects, TC-1 cells (2 × 10^5^ per mouse) were injected subcutaneously into the left flanks of naïve C57BL/6 mice 14 days before immunization. The mice were assigned to groups (6 mice per group) and were immunized with PBS, 10 μg rlipoE7m, 10 μg rlipoE7m plus 10 μg POCpG, and 10 μg rlipoE7m plus 10 μg POCpG/DOTAP. Tumor size was measured by vernier calipers and monitored 3 times a week until tumor size reached 2000 mm^3^.

### 4.7. Analysis of Tumor-Infiltrating Cells

To analyze TC-1 tumor-infiltrating cells, the tumors grown on C57BL/7 mice for 7 days were inoculated with 2 × 10^5^ TC-1 cells via subcutaneous injection. TC-1 tumors were dissected by scissors and then cut into small pieces <2 mm. Next, the small pieces of tumor were further ground by using a syringe plunger to pass through 70-mm cell strainers to gather single cells. The cells (5 × 10^6^) were washed with 2 mL staining buffer (PBS with 1% FBS and 0.04% sodium azide) and resuspended in 0.5 mL staining buffer. To detect the DC, T cell, and myeloid cell populations, the cells were stained with anti-CD8 (53-6.7, FITC; PECy7, BioLegend, San Diego, CA, USA), anti-CD4 (GK1.5, FITC, BD), CD25 (PE, eBioscience), Foxp3 (PECy7, eBioscience), anti-CD45 (EM-05, APC, GeneTex), anti-CD11b (M1/70, PE, eBioscience), anti-Gr-1 (RB6-8C5, PECy7, eBioscience), anti-F4/80 (BM8, FITC, BioLegend), and anti-CD11c (N418, BioLegend) antibodies. Cell populations were analyzed by Attune NxT flow cytometer (Thermo Fisher Scientific, Eugene, OR, USA).

### 4.8. Statistical Analysis

Statistics was analyzed by GraphPad Prism software version 5.02 (GraphPad Software, San Diego, CA, USA). Differences with *p* < 0.05 were considered to be statistically significant. The Kruskal–Wallis test with Dunn’s multiple comparison compared differences for more than two groups. The statistical significance of the tumor growth was analyzed by two-way ANOVA.

## 5. Conclusions

Activating antitumor CTLs and overcoming the effects of tumor-immunosuppressive cells are two goals of cancer immunotherapy. We investigated whether a TLR2 agonist and liposomal TLR9 agonist combination could target DCs in vivo to achieve these two goals. In our previous studies, lipoprotein combined with DOTAP liposomes could target DCs via TLR2 for antigen delivery and stimulate DC maturation [13]. In addition, we also demonstrated that lipoE7m combined with PSCpG could synergistically trigger the TLR2 and TLR9 pathways to decrease tumor immunosuppressive cells, including Tregs and MDSCs, to enhance the therapeutic effect of lipoimmunogens [12]. In this study, using DOTAP liposomes to carry lipoproteins and POCpG was shown to be an achievable approach to develop biocompatible and biodegradable cancer immunotherapeutic vaccines. DOTAP-encapsulated POCpG not only had a prolonged half-life but also an increased delivery efficiency to stimulate intracellular TLR9. Moreover, the DOTAP liposomal combination for lipoprotein and POCpG codelivery via intravenous immunization routes to target cDCs and pDCs augmented IL-12p70 production to promote Th1 responses. Interestingly, rlipoE7m plus POCpG/DOTAP combined could downregulate IL-10 production from both cDCs and pDCs to diminish the number of immunosuppressive Tregs.

## Figures and Tables

**Figure 1 cancers-12-00810-f001:**
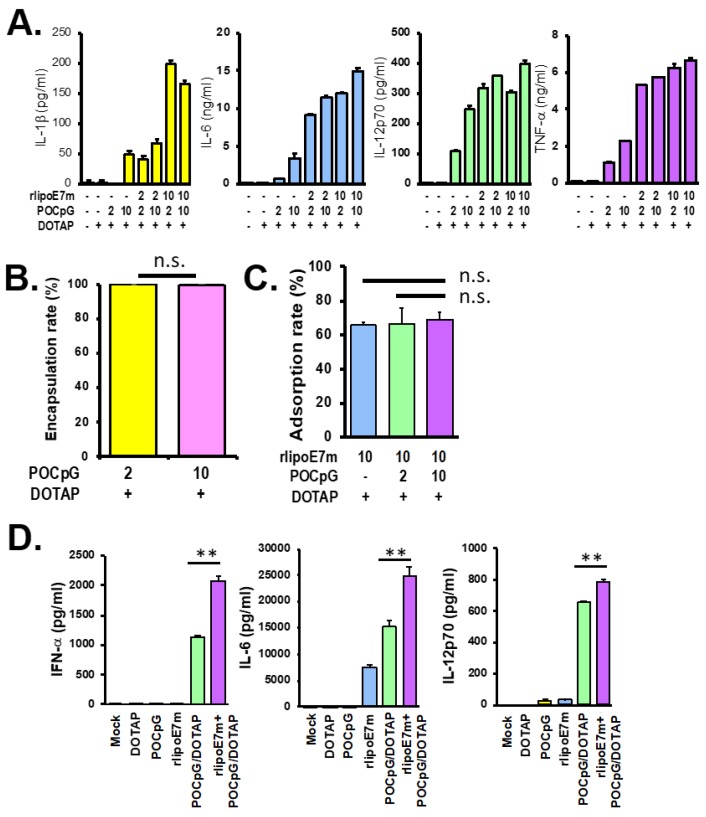
A bone marrow-derived dendritic cells (BMDC)-based bioassay was used to optimize the composition of recombinant lipoprotein with liposomal CpG. (**A**) Two micrograms or 10 μg rlipoE7m and POCpG were mixed with 100 nmole DOTAP to stimulate 2 × 10^6^ BMDCs in 1 mL medium to produce cytokines, such as IL-1β, IL-6, IL-12p70, and TNF-α. After 18 h of incubation, cytokines in the supernatant were determined by ELISA. (**B**) Two micrograms or 10 μg POCpG were encapsulated by 100 nmole DOTAP liposomes. The encapsulation rate was calculated by the formula: (encapsulation rate (%) = (1−free POCpG/loading POCpG) × 100%). (**C**) The recombinant protein rlipoE7m was adsorbed by 100 nmole DOTAP liposomes only, 2 μg POCpG-encapsulated DOTAP liposomes, and 10 μg POCpG-encapsulated DOTAP liposomes. The adsorption rate was calculated by the formula: (adsorption rate (%) = (1−free rlipoE7m in solution / loading rlipoE7m) × 100%). (**D**) A total of 1x10^6^ plasmacytoid dendritic cells (pDCs) cultured from bone marrow cells were incubated with the indicated lipoprotein and liposome combinations for 18 h. Concentrations of IFN-α, IL-6, and IL-12p70 in the supernatant were determined by ELISA. (** *p* < 0.01). The data were collected and analyzed from two independent experiments and n.s. represents no significant difference.

**Figure 2 cancers-12-00810-f002:**
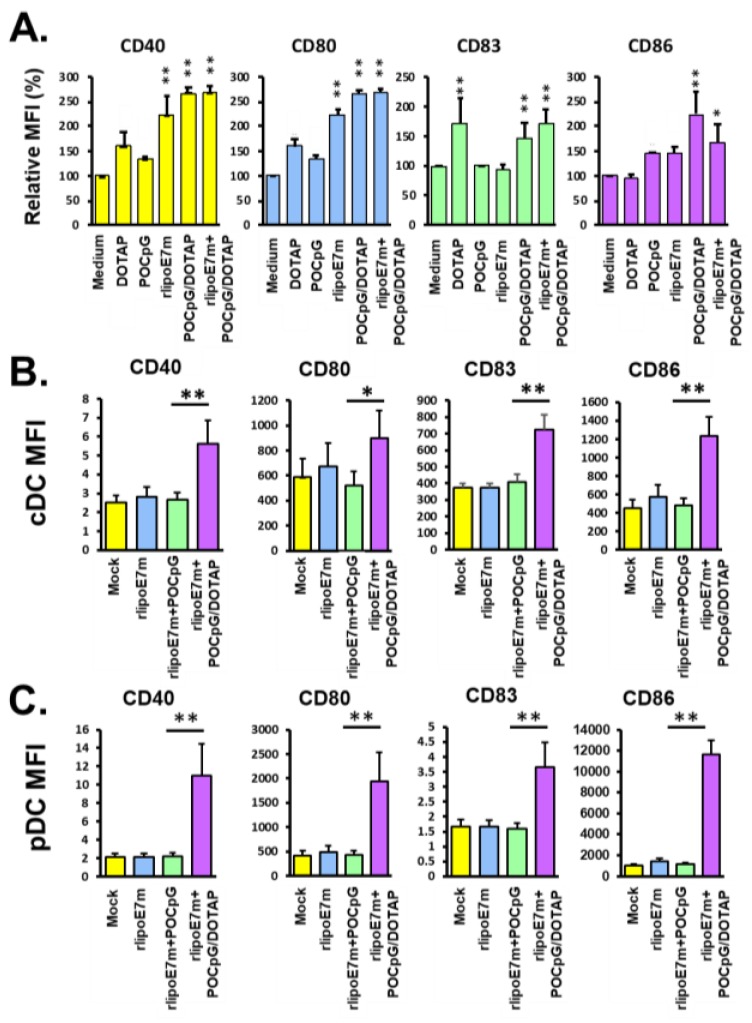
The combination of rlipoE7m and POCpG/DOTAP enhanced the expression of costimulatory molecules on BMDCs in vitro and in vivo. (**A**) Approximately 1 × 10^6^ BMDCs were incubated with 100 nmole DOTAP, 10 μg POCpG, 10 μg rlipoE7m, 10 μg POCpG-encapsulated 100 nmole DOTAP, rlipoE7m, and rlipoE7m plus POCpG/DOTAP combined in 1 mL culture medium for 18 h. The costimulatory molecules CD40, CD80, CD83, and CD86 on BMDCs were analyzed by flow cytometry after gating on CD11c^+^ cells. The relative mean fluorescence intensity (MFI) was calculated by normalizing the MFI of the experimental groups to the MFI of the untreated medium control (each experimental group vs. medium control, * *p* < 0.05, ** *p* < 0.01). (**B** and **C**) C57BL/6 mice were immunized via the tail vein. Conventional dendritic cells (cDCs) and pDCs were collected from spleens, and the indicated surface markers were analyzed by flow cytometry (each mouse group *n* = 6). The MFIs represent the level of surface marker expression. These data were collected and analyzed from three independent experiments. (The rlipoE7m + POCpG/DOTAP group vs. rlipoE7m + POCpG group, * *p* < 0.05, ** *p* < 0.01).

**Figure 3 cancers-12-00810-f003:**
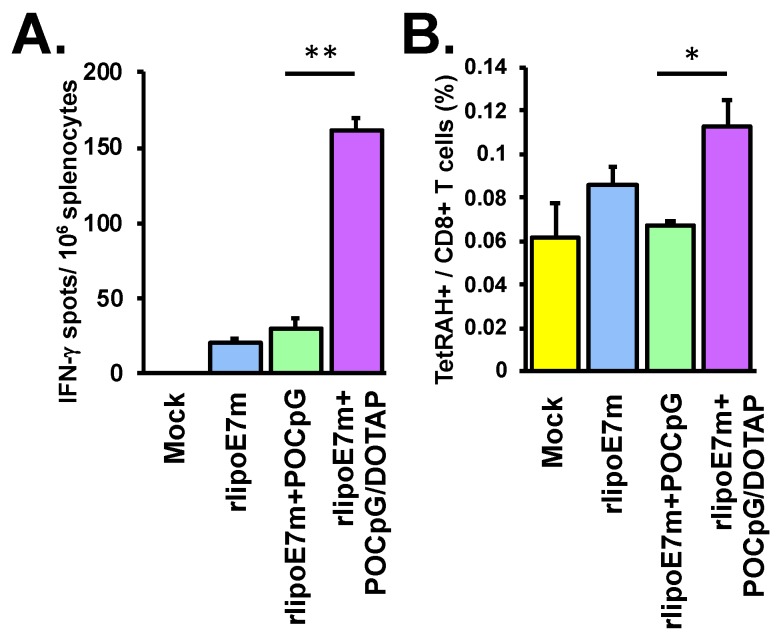
The combination of rlipoE7m and POCpG/DOTAP could elicit cytotoxic T lymphocyte (CTL) responses via tail vein injection. C57BL/6 mice were immunized with rlipoE7m, rlipoE7m + POCpG, or rlipoE7m + POCpG/DOTAP combined on days 0 and 7 via the tail vein. Cells were collected from the spleens of mice that received tail vein injection. (**A**) IFN-γ-producing CTLs were determined by ELISPOT. (**B**) Antigen-specific CTL populations were detected by RAH tetramer and anti-CD8 antibody and then analyzed by flow cytometry. Cells were gated by CD8^+^ staining to determine the percentage of RAH tetramer^+^ cells. These data were collected and analyzed from two independent experiments. (The rlipoE7m + POCpG/DOTAP group vs. rlipoE7m + POCpG group, * *p* < 0.05, ** *p* < 0.01. Each mouse group *n* = 6).

**Figure 4 cancers-12-00810-f004:**
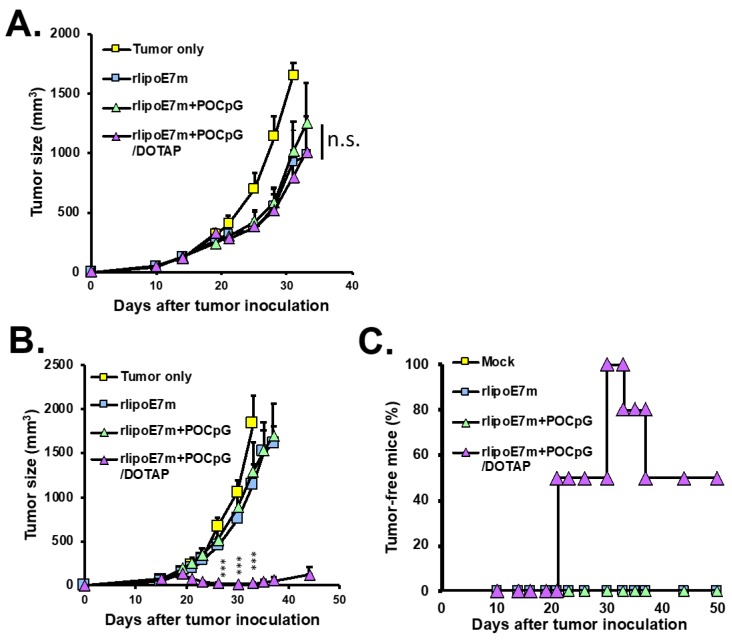
The therapeutic effect of rlipoE7m and POCpG/DOTAP combined via intravenous injection. C57BL/6 mice were inoculated with 2 × 10^5^ TC-1 cells. Fourteen days later, TC-1 tumors were larger than 50 mm^3^, and the tumor-bearing mice were immunized with rlipoE7m, rlipoE7m + POCpG, or rlipoE7m + POCpG/DOTAP combined via tail vein injection. Mice were immunized once (**A**) or twice (**B**) via the tail vein. (**C**) The percentage of tumor-free mice was calculated by the formula (number of tumor-shrinking mice/number of mice per group) × 100%. Tumor size was monitored three times a week. (The rlipoE7m + POCpG/DOTAP group vs. rlipoE7m + POCpG group, *** *p* < 0.001. Each mouse group *n* = 6).

**Figure 5 cancers-12-00810-f005:**
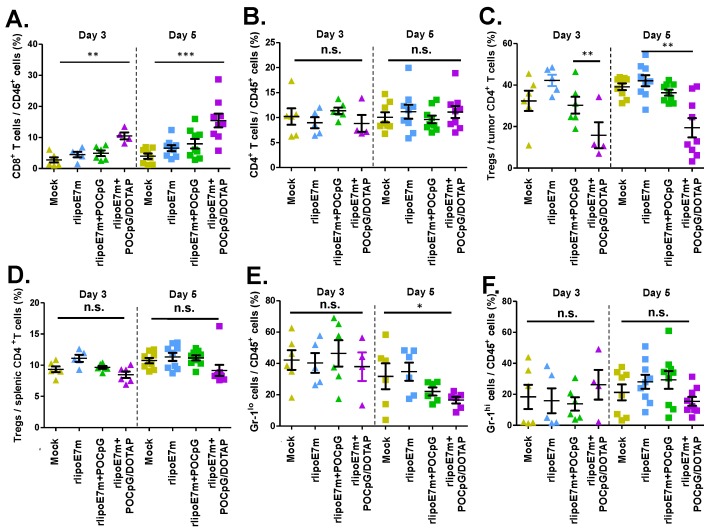
The combination of rlipoE7m plus POCpG/DOTAP could alter tumor-infiltrating immune cells. Mice were inoculated with TC-1 tumor cells and immunized via the tail vein as shown in Figure 4. Tumor-infiltrating cells were collected from tumors 3 and 5 days after the second immunization. The percentage of (**A**) CD8^+^ cells, (**B**) CD4^+^ cells, (**C**) Foxp3^+^ Tregs in CD4^+^ cells, (**E**) monocytic cells and (**F**) granulocytic cells in CD45^+^ tumor-infiltrating cells, and (**D**) Foxp3^+^ Tregs in CD45^+^ cells in spleens were analyzed by flow cytometry. These data were collected and analyzed from two independent experiments. (* *p* < 0.05; ** *p* < 0.01; *** *p* < 0.001. Each mouse group *n* = 5 to 9).

**Figure 6 cancers-12-00810-f006:**
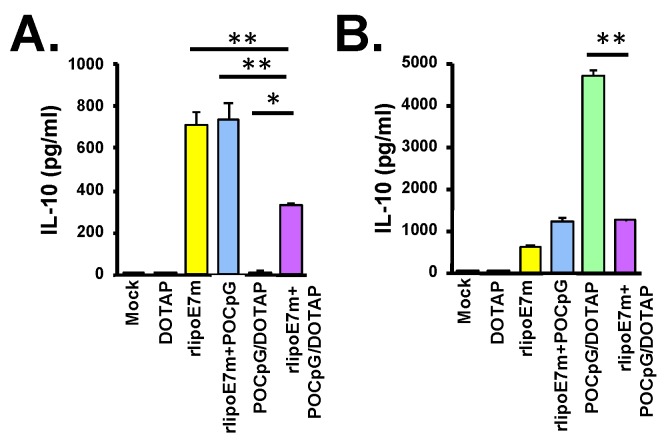
The combination of rlipoE7m and POCpG/DOTAP could decrease IL-10 production from dendritic cells (DCs). (**A**) cDC or (**B**) pDCs cultured from bone marrow-derived cells were incubated with the indicated lipoprotein and liposomal combinations for 24 h. IL-10 in supernatant collected from 1 × 10^6^ cells/mL was determined by ELISA. These data were collected and analyzed from two independent experiments. (* *p* < 0.05; ** *p* < 0.01).

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
