# Peer review of "Liposomal TLR9 Agonist Combined with TLR2 Agonist-Fused Antigen Can Modulate Tumor Microenvironment through Dendritic Cells"

_cancers, 2020, doi:10.3390/cancers12040810_

Round 1
Reviewer 1 Report
A brief summary (one short paragraph):
Thank you for the opportunity to review your manuscript. The manuscript is interesting. I learned a lot from your manuscript. Shen et al. herein reported the development of TLR2 agonist-loaded CpG-ODN-encapsulating cationic liposomes for cancer immunotherapy. There is some concern you need to address as follows.
General comments:
- The title of the manuscript is a bit confusing for me. When I read the title for the first time, I think this manuscript is about the active targeting liposomes against DCs by using a TLR2 ligand at the surface of liposomes, but it is not. So please reconsider the title for clarity.
- Line 28: This sentence is not clear. if I am correct, you injected the rlipoE7m-loaded liposomes via the tail vein. Please re-write this.
- I do not understand why you chose intravenous injection other than i.p. and so on as a route. Please address the reason.
- In this manuscript, you mentioned and illustrated in Fig 1 that the rlipoE7m absorbed- and CpG-ODN encapsulated-DOTAP liposome was prepared. But I think CpG ODN is more likely attached the surface of liposomes, not inside the liposomes. How can you confirmed that CpG ODN is located in the liposomes?
- Figures 2 and 3: Did you check the cell viability treated with the samples you used in the present study?
- What is NP ratios of the complex of CpG ODN and DOTAP liposomes. Besides, did you check the complex formation by gel retardation assay?
- Throughout the manuscript, the student t-test is not appropriate. Please do analyze using one-way ANOVA or two-way ANOVA test.
- In my experience, the liposome composed of only DOTAP is hard to be prepared and quite unstable. Did you check particle size, PDI, and zeta-potential for not only after preparation but also after long term stroage?
- Cationic liposomes will be aggregate when in vitro cell culture in the presence of FBS. Additionally, when the samples proceed with FACS analyses, the aggregated cationic liposomes will appear and be counted in dot plots. Did you carefully check this? Otherwise, the results can not be appropriately interpreted.
- Figures 3 and 5: Did you examined mice that received with DOTAP liposome only, CpG-ODN only, and CpG-ODN plus DOTAP liposome?
- Line 193: "Intravenous injection" is correct? In the materials and methods section, it is "footpad injection".
- You address PS-CpG ODN is not good because of toxic nature; therefore, in this study, PO-CpG ODN is examined. However, in the results section, in vivo toxicity is not investigated. Did you perform in vivo toxicity such as body weight loss?
- Figure 7: Did you examined mice that received with CpG-ODN only?
- Line 339: The CpG-ODN used in this study is class A? Class B? Class C? The nucleotide sequence here is a typo?
- The rlipoE7m is an acidic protein? What is pI?
Minor comments:
- Line 27: "in vivo" and "in vitro" should be italic.
- Line 62: "NOD" should be spelled out.
- Line 67: Please correct "OND" to "ODN".
- Line 69: "DOTAP" should be spelled out.
- Line 78: TNF and IL should be spelled out.
- Line 82: HPV should be spelled out.
- Lines 95-96: BMDCs, CCL, ROS, and MAPK should be spelled out.
- Line 398: TC-1 cells is obtained from ATCC?
Author Response
Reviewer 1:
Thank you for the opportunity to review your manuscript. The manuscript is interesting. I learned a lot from your manuscript. Shen et al. herein reported the development of TLR2 agonist-loaded CpG-ODN-encapsulating cationic liposomes for cancer immunotherapy. There is some concern you need to address as follows.
General comments:
Point 1: The title of the manuscript is a bit confusing for me. When I read the title for the first time, I think this manuscript is about the active targeting liposomes against DCs by using a TLR2 ligand at the surface of liposomes, but it is not. So please reconsider the title for clarity.
Response 1: Thank you very much for the opinions. We change the title as below. “Liposomal TLR9 agonist combined with TLR2 agonist-fused antigen can modulate tumor microenvironment through dendritic cells”.
Point 2: Line 28: This sentence is not clear. if I am correct, you injected the rlipoE7m-loaded liposomes via the tail vein. Please re-write this.
Response 2: Thank you. We re-wrote this. “Combination of rlipoE7m and POCpG/DOTAP could activate conventional DCs and plasmacytoid DCs to augment IL-12 production to promote antitumor responses by intravenous injection.”
Point 3: I do not understand why you chose intravenous injection other than i.p. and so on as a route. Please address the reason.
Response 3: We chose i.v. injection is due to the cationic liposome could be accumulated in tumor site to modulate the tumor microenvironment through enhanced permeability and retention (EPR) effects.
Point 4: In this manuscript, you mentioned and illustrated in Fig 1 that the rlipoE7m absorbed- and CpG-ODN encapsulated-DOTAP liposome was prepared. But I think CpG ODN is more likely attached the surface of liposomes, not inside the liposomes. How can you confirmed that CpG ODN is located in the liposomes?
Response 4: Thank you very much for your comments. Although we did not have direct evidences to show CpG-ODN was encapsulated in DOTAP liposome, we use two different methods to prepare the CpG-ODN/DOTAP liposome to clarify this point. We prepare DOTAP liposome using lipid film method. The PO-CpG ODN was added in water (without salt contained), and then added to DOTAP lipid film to form CpG-ODN encapsulated-DOTAP. On the other hand, we mixed PO-CpG ODN to DOTAP liposome to form CpG-ODN adsorbed-DOTAP. We tested these two kinds of liposomes to stimulated BMDCs to secrete TNF-a. We observed that CpG-ODN encapsulated-DOTAP could induce higher levels of TNF-a secretion than CpG-ODN adsorbed-DOTAP (Fig. S1A). Furthermore, we immunized these two kinds of liposomes plus lipoE7m into B6 mice to induce CTL responses. The results showed that CpG-ODN encapsulated-DOTAP immunization could elicit stronger CTL responses than CpG-ODN adsorbed-DOTAP immunization (Fig. S1B). Therefore, we suggested that CpG-ODN encapsulated-DOTAP could be delivered into endosomes to trigger TLR9 signaling efficiently.
For reviewer
Figure S1. The CpG encapsulated DOTAP could enhance TNF-a secretion from BMDCs and CTL responses. (A) A total of 1x106 BMDCs cultured from bone marrow cells were incubated with the indicated lipoprotein, POCpG and liposome combinations for 18 h. Concentrations of TNF-a in the supernatant were determined by ELISA. (B) C57BL/6 mice were immunized with rlipoE7m, rlipoE7m + POCpG and rlipoE7m + POCpG/DOTAP combined on days 0 and 7 via the tail vein. IFN-g-producing CTLs were determined by ELISPOT.
Point 5: Figures 2 and 3: Did you check the cell viability treated with the samples you used in the present study?
Response 5: We have checked the cell viability [J Control Release. 2016 Jul 10;233:57-63.]. We found that 300 nmole DOTAP could induce cell apoptosis. Thus, we used 100 nmole DOTAP to do our experiments.
Point 6: What is NP ratios of the complex of CpG ODN and DOTAP liposomes. Besides, did you check the complex formation by gel retardation assay?
Response 6: Given 1 mole N per mole DOTAP and 19 phosphate groups in CpG-ODN 1826, N/P ratio of 10 ug CpG/ 100nmole DOTAP is 3.35. We did not check the complexes by gel retardation. Alternatively, we used viva spin 500 to collect the filtration of CpG-DOTAP complex to determine the amount of free CpG (Fig. 1B). The result showed that CpG and DOTAP was 99.7% combination.
Point 7: Throughout the manuscript, the student t-test is not appropriate. Please do analyze using one-way ANOVA or two-way ANOVA test.
Response 7: Thanks for your comments. We used ANOVA test to analyze data in all figures.
Point 8: In my experience, the liposome composed of only DOTAP is hard to be prepared and quite unstable. Did you check particle size, PDI, and zeta-potential for not only after preparation but also after long term stroage?
Response 8: This information was published in our previous paper [J Control Release. 2016 Jul 10;233:57-63.]. DOTAP particle size: 100 nm, DOTAP zeta-potential: 35-40 mV. We have been monitored the stability for 14 days.
Point 9: Cationic liposomes will be aggregate when in vitro cell culture in the presence of FBS. Additionally, when the samples proceed with FACS analyses, the aggregated cationic liposomes will appear and be counted in dot plots. Did you carefully check this?
Otherwise, the results can not be appropriately interpreted.
Response 9: Thanks for your comments. We did not see obvious liposome aggregates during cell culture. Because we stained cells using anti-CD11c to gate dendritic cells population, the potential aggregate cannot be gated for further analysis.
Point 10: Figures 3 and 5: Did you examined mice that received with DOTAP liposome only, CpG-ODN only, and CpG-ODN plus DOTAP liposome?
Response 10: At the beginning, we tested DOTAP liposome and CpG-ODN plus DOTAP via s.c. injection. DOTAP liposome alone could not inhibit tumor growth (J Control Release. 2016 Jul 10;233:62 Fig. 5A). Similarly, CpG-ODN plus DOTAP could not inhibit tumor growth (Fig. S2) or elicit CTL responses (Fig. S1B).
For reviewer:
Figure S2. The therapeutic effect of rlipoE7m and POCpG/DOTAP was via footpad immunization. C57BL/6 mice were inoculated with 2×105 TC-1 cells. 14 days later, TC-1 tumors were bigger than 50 mm3 and then the tumor-bearing mice were immunized with 10 mg rlipoE7m, 10 mg POCpG/DOTAP via tail vein injection.
Point 11: Line 193: "Intravenous injection" is correct? In the materials and methods section, it is "footpad injection".
Response 11: Thank you. We corrected that in Line 390.
Point 12: You address PS-CpG ODN is not good because of toxic nature; therefore, in this study, PO-CpG ODN is examined. However, in the results section, in vivo toxicity is not investigated. Did you perform in vivo toxicity such as body weight loss?
Response 12: We did not investigate the potential toxicity of PO-CpG, because the toxicity may be not acute toxicity but long-term effects. The PS-CpG is not easy to be digested in vivo. In contrast, the native PO-CpG is easy to be digested in vivo. We observed that PO-CpG plus rlipoE7m immunization could not elicit CTL responses (Fig. 4A and B) or inhibit tumor growth as well as combination of rlipoE7m and PO-CpG/DOTAP liposome.
Point 13: Figure 7: Did you examined mice that received with CpG-ODN only?
Response 13: We test lipoE7m plus PO-CpG which did not affect the tumor growth.
Point 14: Line 339: The CpG-ODN used in this study is class A? Class B? Class C? The nucleotide sequence here is a typo?
Response 14: The CpG-ODN 1826, a class B type, is used in this study and purchased from InvivoGen. We corrected the sequence: 5’- TCCATGAGCTTCCTGAGCTT -3’
Point 15: The rlipoE7m is an acidic protein? What is pI?
Response 15: The rlipoE7m is an acidic protein because of a 18.62% acidic amino acids composition. And the pI value of rlipoE7m is 4.82 by vector NTI calculation. Also, the rlipoE7m is negatively charged at pH7.
Minor comments:
- Line 27: "in vivo" and "in vitro" should be italic.
- Line 62: "NOD" should be spelled out.
- Line 67: Please correct "OND" to "ODN".
- Line 69: "DOTAP" should be spelled out.
- Line 78: TNF and IL should be spelled out.
- Line 82: HPV should be spelled out.
- Lines 95-96: BMDCs, CCL, ROS, and MAPK should be spelled out.
Response 1-7: Thank you. We corrected these.
- Line 398: TC-1 cells is obtained from ATCC?
Response 8: The TC-1 cell line, which expresses HPVE6 and E7 mouse epithelial lung cancer, was a kind gift from Dr. T-C. Wu (Johns Hopkins University, USA).
Reviewer 2 Report
I have now completed reviewing the article “Liposomal TLR9 agonist combined with TLR2 agonist-fused antigen can target dendritic cells to modulate tumor microenvironment”. This is a very interesting study that was well described overall. There were no major concerns in the manuscript, only some minor corrections needed before the article is suitable for publication.
- Because many abbreviations were used in current manuscript, the reviewer suggests that add an Abbreviation section, and put those abbreviations into the abbreviations section.
- Discussion section: Shortcoming and/or future work are not well addressed. These should be well spelled out and discussed.
Author Response
Reviewer 2:
Comments and Suggestions for Authors
I have now completed reviewing the article “Liposomal TLR9 agonist combined with TLR2 agonist-fused antigen can target dendritic cells to modulate tumor microenvironment”. This is a very interesting study that was well described overall. There were no major concerns in the manuscript, only some minor corrections needed before the article is suitable for publication.
Point 1: Because many abbreviations were used in current manuscript, the reviewer suggests that add an Abbreviation section, and put those abbreviations into the abbreviations section.
Response 1: Thank your opinions. We added an abbreviation section.
Point 2: Discussion section: Shortcoming and/or future work are not well addressed.
Response 2: Thank you. We added a paragraph.
“Liposomes are effective delivery system for nano drug and nucleic acids in clinical application. The cationic DOTAP liposome can adsorb or encapsulate RNA and DNA for genetic transfection [42]. Liposomes are a feasible approach for intravenous and intramuscular injection into circulation system to induce humoral and cellular immunity [43-45]. To turn cold tumors into hot tumors, immunotherapeutic agents should be delivered into the tumor site to modulate immunosuppressive microenvironment [46,47]. In this report, we demonstrated that DOTAP liposome could carry TLR2-fused antigen and TLR9 agonist to enhance DCs activation and reduce Tregs in tumor microenvironment. Our study could provide some idea to design immunostimulators or modulators delivery for cancer immunotherapy through regulation of tumor microenvironment.”
Reference:
- Ewert, K.K.; Zidovska, A.; Ahmad, A.; Bouxsein, N.F.; Evans, H.M.; McAllister, C.S.; Samuel, C.E.; Safinya, C.R. Cationic liposome-nucleic acid complexes for gene delivery and silencing: pathways and mechanisms for plasmid DNA and siRNA. Top Curr Chem 2010, 296, 191-226, doi:10.1007/128_2010_70.
- Reidel, I.G.; Camussone, C.; Suarez Archilla, G.A.; Calvinho, L.F.; Veaute, C. Liposomal and CpG-ODN formulation elicits strong humoral immune responses to recombinant Staphylococcus aureus antigens in heifer calves. Vet Immunol Immunopathol 2019, 212, 1-8, doi:10.1016/j.vetimm.2019.04.011.
- Mansury, D.; Ghazvini, K.; Amel Jamehdar, S.; Badiee, A.; Tafaghodi, M.; Nikpoor, A.R.; Amini, Y.; Jaafari, M.R. Increasing Cellular Immune Response in Liposomal Formulations of DOTAP Encapsulated by Fusion Protein Hspx, PPE44, And Esxv, as a Potential Tuberculosis Vaccine Candidate. Rep Biochem Mol Biol 2019, 7, 156-166.
- Bulbake, U.; Doppalapudi, S.; Kommineni, N.; Khan, W. Liposomal Formulations in Clinical Use: An Updated Review. Pharmaceutics 2017, 9, doi:10.3390/pharmaceutics9020012.
- Li, J.; Byrne, K.T.; Yan, F.; Yamazoe, T.; Chen, Z.; Baslan, T.; Richman, L.P.; Lin, J.H.; Sun, Y.H.; Rech, A.J., et al. Tumor Cell-Intrinsic Factors Underlie Heterogeneity of Immune Cell Infiltration and Response to Immunotherapy. Immunity 2018, 49, 178-193 e177, doi:10.1016/j.immuni.2018.06.006.
- Rodallec, A.; Sicard, G.; Fanciullino, R.; Benzekry, S.; Lacarelle, B.; Milano, G.; Ciccolini, J. Turning cold tumors into hot tumors: harnessing the potential of tumor immunity using nanoparticles. Expert Opin Drug Metab Toxicol 2018, 14, 1139-1147, doi:10.1080/17425255.2018.1540588.
Round 2
Reviewer 1 Report
I have gone through the revised manuscript, and I think all my comments have been adequately addressed except the author’s response 14.
The authors mentioned that the used CpG-ODN is class B 1825 type. Firstly, the sequence in the materials and methods section is still wrong. Besides, I do not understand the reason why CpG-ODN 1826 was used in this study since CpG ODN is well known potent inducer for B cells with relatively weak inducer for Th1 responses. I would like to make this clear.
Author Response
Comments and Suggestions for Authors
I have gone through the revised manuscript, and I think all my comments have been adequately addressed except the author’s response 14.
The authors mentioned that the used CpG-ODN is class B 1825 type. Firstly, the sequence in the materials and methods section is still wrong. Besides, I do not understand the reason why CpG-ODN 1826 was used in this study since CpG ODN is well known potent inducer for B cells with relatively weak inducer for Th1 responses. I would like to make this clear.
Response: Thanks for your comments. I am very sorry for the mistake. We have corrected it at Line 404: The POCpG sequence is 5’- TCCATGACGTTCCTGACGTT -3’ (GeneDireX, Nevada, USA).
Although the class B type CpG-ODN has been reported that can stimulate B cells and NK cells, immunization class B type CpG-ODN with antigens still can induce very strong Th1-biased immune responses (Chu et al. J. Exp. Med., 186(10): 1623–1631, 1997; Switaj et al. Clin. Cancer Res. 10:4165-75, 2004; Davis et al. J Immunol. 160: 870-876, 1998). In our previous reports, we observed that the class B type CpG-ODN could stimulate dendritic cell to secrete Th1 cytokine, IL-12 and promote antitumor CTL response (J Control Release. 173: 158, 2014; Scientific Reports 5: 12578, 2015). That’s why we used the class B type CpG-ODN in our studies.
Reference:
- Chu RS, Targoni OS, Krieg AM, Lehmann PV, Harding CV. CpG oligodeoxynucleotides act as adjuvants that switch on T helper 1 (Th1) immunity.
- Switaj T, Jalili A, Jakubowska AB, Drela N, Stoksik M, Nowis D, Basak G, Golab J, Wysocki PJ, Mackiewicz A, Sasor A, Socha K, Jakóbisiak M, Lasek W. CpG immunostimulatory oligodeoxynucleotide 1826 enhances antitumor effect of interleukin 12 gene-modified tumor vaccine in a melanoma model in mice. Clin Cancer Res. 2004 Jun 15;10(12 Pt 1):4165-75.
- Davis HL, Weeratna R, Waldschmidt TJ, Tygrett L, Schorr J, Krieg AM. CpG DNA is a potent enhancer of specific immunity in mice immunized with recombinant hepatitis B surface antigen. J Immunol. 1998 Jan 15;160(2):870-6
- Song YC, Cheng HY, Leng CH, Chiang SK, Lin CW, Chong P, Huang MH, Liu SJ. A novel emulsion-type adjuvant containing CpG oligodeoxynucleotides enhances CD8+ T-cell-mediated anti-tumor immunity. J Control Release. 2014 Jan 10;173:158-65.
- Song YC, Liu SJ. A TLR9 agonist enhances the anti-tumor immunity of peptide and lipopeptide vaccines via different mechanisms. Sci Rep. 2015 Jul 28;5:12578.